# The Iridoid Glycoside Loganin Modulates Autophagic Flux Following Chronic Constriction Injury-Induced Neuropathic Pain

**DOI:** 10.3390/ijms232415873

**Published:** 2022-12-14

**Authors:** Kuang-I Cheng, Yu-Chin Chang, Li-Wen Chu, Su-Ling Hsieh, Li-Mei An, Zen-Kong Dai, Bin-Nan Wu

**Affiliations:** 1Department of Anesthesiology, School of Medicine, College of Medicine, Kaohsiung Medical University, Kaohsiung 80708, Taiwan; 2Department of Anesthesiology, Kaohsiung Medical University Hospital, Kaohsiung 80708, Taiwan; 3Department of Pharmacology, Graduate Institute of Medicine, College of Medicine, Drug Development and Value Creation Research Center, Kaohsiung Medical University, Kaohsiung 80708, Taiwan; 4Department of Cosmetic Application and Management, Department of Nursing, Yuh-Ing Junior College of Health Care and Management, Kaohsiung 80776, Taiwan; 5Department of Pharmacy, Kaohsiung Medical University Hospital, Kaohsiung 80708, Taiwan; 6Department of Pediatrics, School of Medicine, College of Medicine, Kaohsiung Medical University, Kaohsiung 80708, Taiwan; 7Department of Pediatrics, Division of Pediatric Cardiology and Pulmonology, Kaohsiung Medical University Hospital, Kaohsiung 80708, Taiwan; 8Department of Medical Research, Kaohsiung Medical University Hospital, Kaohsiung 80708, Taiwan

**Keywords:** peripheral nerve injury, neuropathic pain, loganin, autophagy, apoptosis

## Abstract

Autophagy facilitates the degradation of organelles and cytoplasmic proteins in a lysosome-dependent manner. It also plays a crucial role in cell damage. Whether loganin affects autophagy in chronic constriction injury (CCI)-induced neuropathic pain remains unclear. We investigated the neuroprotective effect of loganin on the autophagic–lysosomal pathway in the rat CCI model. Sprague–Dawley rats were divided into sham, CCI, sham + loganin, and CCI + loganin. Loganin (5 mg/kg/day) was intraperitoneally injected once daily, and rats were sacrificed on day 7 after CCI. This study focused on the mechanism by which loganin modulates autophagic flux after CCI. CCI enhanced the autophagic marker LC3B-II in the ipsilateral spinal cord. The ubiquitin-binding protein p62 binds to LC3B-II and integrates into autophagosomes, which are degraded by autophagy. CCI caused the accumulation of p62, indicating the interruption of autophagosome turnover. Loganin significantly attenuated the expression of Beclin-1, LC3B-II, and p62. Double immunofluorescence staining was used to confirm that LC3B-II and p62 were reduced by loganin in the spinal microglia and astrocytes. Loganin also lessened the CCI-increased colocalization of both proteins. Enhanced lysosome-associated membrane protein 2 (LAMP2) and pro-cathepsin D (pro-CTSD) in CCI rats were also attenuated by loganin, suggesting that loganin improves impaired lysosomal function and autophagic flux. Loganin also attenuated the CCI-increased apoptosis protein Bax and cleaved caspase-3. Loganin prevents CCI-induced neuropathic pain, which could be attributed to the regulation of neuroinflammation, neuronal autophagy, and associated cell death. These data suggest autophagy could be a potential target for preventing neuropathic pain.

## 1. Introduction

Neuropathic pain, which involves neuroinflammation in the spinal cord, is a severely debilitating condition that adversely affects the quality of life. Various nerve injury stimuli in the peripheral or central nervous system can result in neuropathic pain with characteristic spontaneous pain, hyperalgesia, and allodynia [1,2]. Suitable therapies to attenuate neuropathic pain are limited because the mechanism of pathophysiology causing this is not completely understood.

Loganin is an iridoid glycoside, the main active ingredient extracted from Cornus officinalis, which possesses antioxidant, anti-inflammatory, and anti-diabetic properties [3,4,5]. Numerous reports have exhibited that loganin has various neuroprotective effects. Loganin attenuated neuroinflammatory responses by inactivating NF-κB in Aβ25-35-induced PC12 cells [6]. In a Parkinson’s disease mouse model, loganin exerted neuroprotective effects by reducing inflammation and apoptosis [7]. Loganin also attenuated Aβ1–42-mediated inflammation in microglia-like BV2 cells by regulating the TLR4/TRAF6/NFκB pathway [8]. We have shown that loganin improved chronic constriction injury (CCI)-mediated neuropathic pain behaviors and reduced proinflammatory cytokines and inflammatory proteins in rat sciatic nerves [9]. Loganin also repressed CXCL12/CXCR4-mediated neuropathic pain through the NOD-like receptor protein 3 (NLRP3) inflammasome in CCI rats [10]. However, limited information has been available on how loganin affects CCI-induced neuropathic pain.

Autophagy involves pathological conditions, including inflammatory diseases, cancer, and neurodegenerative diseases [11]. It maintains cellular homeostasis by clearing damaged organelles and proteins [12]. Research has also shown that autophagy regulates neuropathic pain. One report found that autophagy in GABAergic interneurons affects neuropathic pain [13]. In contrast, another showed that autophagy modulates neuropathic pain via the mammalian target of the rapamycin (mTOR) signaling pathway [14]. Previous articles demonstrated that autophagy is blocked in the spinal dorsal horn after CCI or spinal nerve injury, which can trigger neuropathic pain [15,16]. Based on those findings, we hypothesized that autophagy has an essential role in the maintenance of neuropathic pain. Modulation of autophagy may be a potential therapeutic target in human diseases.

The pathogenesis of neuropathic pain is not entirely understood. We further aimed to investigate the possible effects of loganin on neuropathic pain-associated autophagy using the CCI rat model. We found that autophagy impairment in the spinal cord led to an enhancement of neuropathic pain. In contrast, pharmacological agents stimulating autophagy could reduce inflammation and sustained pain. In this study, we suggest that loganin might improve neuropathic pain by stimulating autophagy and inhibiting apoptosis in the spinal cord of CCI rats.

## 2. Results

### 2.1. Loganin Promoted Autophagy in CCI Rats

Since the effects of loganin on pain behaviors have been published previously [10], these specimens were from the same batch of rats to reduce unnecessary sacrifices and ensure ethical treatments. To explore the mechanisms of autophagy after CCI, we evaluated the expression of autophagy in the spinal cord proteins by Western blot. The regulatory protein Beclin1 is involved in initiating autophagy and autophagosome formation [17]. Beclin-1 levels were significantly elevated on day 7 after CCI, compared with the sham group (Figure 1A,B), suggesting that autophagy is activated after CCI.

The conversion of LC3B-I protein into LC3B-II is a reliable marker of the autophagy mechanism [18]. LC3B-II levels increased on day 7 after CCI compared with the sham group (Figure 1A,B). However, enhanced LC3B-II levels can reflect either the inhibition of autophagy or the failure to clean up the autophagosomes. The adaptor protein p62/SQSTM1 delivers ubiquitinated cargo to autophagosomes and is degraded by autophagy [19]. Levels of p62 significantly increased on day 7 after CCI and reverted in the loganin-treated group (Figure 1A,B). Loganin reduced p62 to a level comparable to that of the sham group. These findings were consistent with the abnormal p62 protein degradation after CCI, including decreased autophagic flux.

### 2.2. Loganin-Induced Autophagy Occurred in Neuronal and Microglial Cells in CCI Rats

Autophagy is the dynamic process of sequestering organelles or cytoplasmic proteins into lytic components. To further investigate the cell-type specificity of autophagosome accumulation, we co-immunostained the spinal cord sections with LC3B and different cell-type markers (Figure 2A–C). The total number of LC3B-positive cells significantly increased in the spinal dorsal horn after CCI (Figure 2D). In the spinal dorsal horn, LC3B showed a high level of colocalization with NeuN (neuron marker) and Iba1 (microglia marker) but not GFAP (astrocyte marker) in the CCI group. Those effects were attenuated in the loganin-treated group (Figure 2E). These results indicated that loganin improved the impairment of autophagy in neurons and microglia but not astrocytes.

### 2.3. Autophagosomes Accumulate after CCI Because of Impaired Autophagic Flux

Disruption of autophagic flux is related to neuropathic pain [15]. To investigate autophagic flux, it is necessary to determine the degradation of the ubiquitin-binding protein p62/SQSTM1, an autophagic substrate, to monitor autophagic activity. Since p62 is degraded by autophagy along with its cargo, the accumulation of p62 suggests disruption of autophagic degradation. CCI promotes ubiquitin- and LC3B-binding protein p62 accumulation in the ipsilateral spinal cord. This result inversely correlates with autophagic activity, indicating the blockage of autophagosome turnover (Figure 3A,B). CCI increased the colocalization of LC3B and p62, and the effect was inhibited by loganin (Figure 3A,C). Accumulation of autophagosomes in CCI rats could be attributed to the impairment of autophagic flux. This impaired autophagy can be reversed by loganin. The pathological accumulation of autophagosomes is predominant in neurons associated with signs of impaired lysosomes.

### 2.4. Lysosomal Dysfunction Contributed to the Disruption of Autophagic Flux

Lysosomes fuse with autophagosomes to permit the degradation of autophagosomal cargo by lysosomal hydrolases [20]. The lysosome-associated membrane protein 2 (LAMP2) was immunostained to estimate the lysosomal defects after CCI. We found increased LAMP2 in CCI rats, which was significantly attenuated on day 7 after loganin treatment (Figure 4A,B). CCI also exhibited increases in LC3B-II, indicating autophagosome cleaning failure. Furthermore, cathepsin D (CTSD) is an essential protease regulating lysosome proteolytic activity in neurons [21]. Accumulating immature CTSD forms (pro-CTSD) is used as a marker of lysosomal impairment. The expression of the pro-CTSD protein was increased in CCI rats, which was decreased following loganin treatment (Figure 4C,D). Altogether, these findings suggested that loganin prevents CCI-induced lysosomal dysfunction and autophagosome accumulation.

### 2.5. Autophagic Flux Disruption Is Associated with Neuronal Cell Death

We next tested whether the disruption of autophagosome degradation caused neuronal cell death after CCI. The expression of apoptosis-related proteins (Bax and cleaved caspase3) was measured. Loganin significantly diminished the increasing cleaved caspase-3 and proapoptotic protein Bax levels caused by CCI (Figure 5A,B). We used double immunofluorescence staining to characterize the aforementioned defects for cleaved caspase3 and NeuN (neuronal marker). We observed that cleaved caspase3 was greatly increased and colocalized with neurons after CCI in the spinal dorsal horn (Figure 5C–E). In particular, loganin treatment significantly prevented CCI-induced apoptosis.

## 3. Discussion

This study shows the correlation between autophagy, analgesic effects, and the anti-neuroinflammation effects [10] of loganin on CCI-induced neuropathic pain. Loganin significantly affected the increases of Beclin-1, LC3B-II, p62, LAMP2, and pro-CTSD in the ipsilateral spinal cord of CCI rats. In the CCI group, increased LC3B-II was paralleled by p62 accumulation in the spinal cord and spinal glial cells, suggesting a block in autophagic flux. Loganin also reduced the CCI-enhanced apoptosis-related proteins (Bax and cleaved caspase-3). These results indicated that loganin’s analgesic and anti-neuroinflammatory effects and neuroprotection could be attributed in part to facilitating autophagic flux in rats with CCI-induced neuropathic pain.

The pathophysiological mechanism of neuropathic pain is complex. Autophagy is described by the formation of autophagosomes, which fuse with the lysosome to deliver the contents into the autolysosomes, where they are degraded [12]. Although the precise mechanism of autophagy’s contribution to neuropathic pain is not well understood, several neuropathic pain models have demonstrated changes in autophagic flux. Autophagy was disrupted in the mice’s spinal cords that underwent spinal nerve ligation (SNL). The accumulation of the ubiquitin- and LC3-binding protein p62 likely results from a block of autophagosome turnover [15]. In spare nerve injury (SNI) models, dysfunction of spinal autophagy could be relevant for altered nociceptive processing [22]. Enhancing autophagy through pharmacological approaches can slow neuropathic pain [23]. Conversion of LC3B-l to LC3B-ll by adding phosphatidylethanolamine (PE) is essential for the formation of autophagosomes and is a marker of autophagosome formation and accumulation. Moreover, the adaptor protein p62/SQSTM1 mediates the transport of ubiquitinated cargo to autophagosomes. Accumulation of p62 inversely correlates with autophagic activity. Our results show that CCI leads to LC3B-ll accumulation in neurons and a block of autophagic flux. Loganin decreases the colocalization of LC3B-II and p62, indicating that loganin promotes autophagic flux in neurons. Recently, autophagy was shown to be a cell-protective mechanism after injury under various circumstances, including oxidative stress [24], inflammatory response [25], acute liver injury [26], and cerebral ischemia [27]. Induced by various stresses, autophagy is upregulated to participate in cell adaption and survival. However, when the autophagic flux is impaired, damaged organelles accumulate, and protein aggregates interfere with cellular functions and may eventually result in cell death.

Elevated autophagic flux appears because of autophagosome degradation. This process requires the fusion of autophagosomes and lysosomes [28]. Our results suggested that autophagic flux is significantly disturbed after CCI. We then examined whether the lysosomal function has significance for CCI-induced neuropathic pain. LAMP2 is a heavily glycosylated membrane protein protecting lysosomal membranes from self-digestion and CTSD is a necessary protease that regulates lysosome proteolytic activity. Lysosome integrity is affected by disruption in disease states [29]. We found impaired lysosomal protease on day 7 after CCI. However, autophagy can also contribute to cell death when the lysosomal function is compromised. We then sought to understand the possible interplay between apoptosis and autophagy. Caspase3, the major executioner caspase, is cleaved to an active form in apoptosis. This study shows that CCI-induced lysosomal defects resulting from autophagosome accumulation in neurons occur when autophagy is blocked, which likely contributes to neuronal cell death.

In our previous report, CCI-enhanced mechanical allodynia and thermal hyperalgesia were significantly inhibited by loganin. Loganin also decreased proinflammatory cytokine levels (such as TNF-α and IL-1β) by reducing astrocyte and microglia activation [9,10]. The induction of microglial autophagy decreases cytokine production by inhibiting inflammasome formation, which could reduce neuropathic pain [30]. Our findings extend previous findings in the literature by demonstrating that loganin alleviated neuropathic pain after CCI and restored autophagic flux.

Overall, our data provide evidence that loganin could be an effective agent following peripheral nerve injury, enhancing cell-protective autophagy while attenuating neuropathic pain. These results suggest that the regulation of autophagy opens novel therapeutic avenues to improve the patient’s quality of life and reduce suffering from neuropathic pain. Revealing novel cellular and molecular mechanisms behind neuroinflammation and neuropathic pain are indispensable for developing potential targets of pharmacotherapy.

## 4. Materials and Methods

### 4.1. Experimental Animals

Six-week-old male Sprague-Dawley (SD) rats (*n* = 72) (BioLASCO Taiwan Co., Ltd., Taipei, Taiwan) were used in all experiments. Animals were adapted to standard laboratory conditions for at least one week, and food and water were available ad libitum. They were divided into sham, CCI, sham + loganin, and CCI + loganin groups [10]. All protocols were approved by the Animal Care and Use Committee (IACUC approval No. 105112, 1 August 2017 to 31 July 2020) of Kaohsiung Medical University and adhered to the National Institute of Health guidelines for the use of the experimental animals.

### 4.2. Drugs

Loganin (purity > 98%, HPLC) was obtained from Abcam (ab143653; Cambridge, UK) and dissolved in sterile physiological saline (0.9% NaCl). Animals received daily intraperitoneal injections of loganin at 5 mg/kg (sham + loganin, CCI + loganin) [9,10], and they were sacrificed on day 7 after surgery.

### 4.3. CCI Model Preparation

The CCI model is considered an appropriate model of neuropathic pain [31]. Unilateral CCI surgery was performed as described [31,32,33,34], and the ipsilateral spinal cord was collected for the following experiments. Briefly, SD rats were anesthetized with pentobarbital sodium (40 mg/kg) intraperitoneally. The sciatic nerve was exposed and freed, and then 3 ligatures of 4-0 chromic catgut were placed around the sciatic nerve [32,33,34,35]. Three ligatures were tied loosely to the rat’s hind limb, causing a brief twitch. The same surgical procedures were executed in sham-operated rats, but no ligatures were placed.

### 4.4. Western Blot

Animals were sacrificed on day 7 after CCI surgery, and the lumbar spinal segment (L4-L5) was rapidly removed. Total protein in the ipsilateral part of the rats’ spinal cord was extracted. The protein was separated by sodium dodecyl sulfate–polyacrylamide gel electrophoresis (SDS-PAGE) and transferred to a PVDF membrane. After blocking, the membranes were incubated overnight (4 °C) with primary antibody against Beclin 1 (1:1000, #3495; Cell Signaling, Danvers, MA, USA), p62 (1:1000, ab56416; Abcam, Cambridge, UK), LC3B (1:1000, L7543; Sigma-Aldrich, St. Louis, MO, USA), Cathepsin D (1:1000, MA5-17236; Thermo Fisher Scientific, Waltham, MA, USA), Bax (1:1000, #2772; Cell Signaling), cleaved caspase3 (1:1000, #9661S; Cell Signaling) and β-actin (1:5000, A5441; Sigma-Aldrich). After washing, membranes were incubated with the corresponding HRP-conjugate IgG secondary antibody. The target proteins’ bands were visualized using ECL Western blotting detection reagents (#WBKLS0500; Millipore, Temecula, CA, USA).

### 4.5. Immunofluorescence

SD rats were anesthetized, and the ipsilateral spinal cord was removed on day 7 after surgery. Sample sections were incubated overnight with primary antibody against NeuN-Cy3 (1:100, ABN78C3; Millipore, Temecula, CA, USA), Iba1 (1:100, ab15690; Abcam, Cambridge, UK), GFAP (1:100; Sigma-Aldrich, St. Louis, MO, USA), p62 (1:100, ab56416; Abcam), LC3B (1:100, L7543; Sigma-Aldrich), LAMP2 (1:50, PA1-655; Thermo Fisher Scientific, Waltham, MA, USA), and cleaved caspase3 (1:100, #9661S; Cell Signaling). After washing, the sections were incubated in the secondary antibody for 2 h at room temperature. Each sample section was stained with DAPI, followed by coverslips, and mounted onto slides. All samples were mounted with Fluoroshield™ with DAPI (F6057; Sigma-Aldrich; blue-fluorescent dye) and analyzed on a Zeiss LSM700 (Carl Zeiss, Oberkochen, Germany) confocal microscope with Zen software. Images were processed and quantified using Photoshop (CS6; Adobe) and ImageJ (version 1.53j; National Institutes of Health, Bethesda, MD, USA).

### 4.6. Statistical Analysis

Experimental data in bar graphs were expressed as the mean ± SE (*n* = 6) with each experiment conducted in triplicate. Pearson’s correlation coefficients are represented for the colocalization analysis. One-way analysis of variance (ANOVA) was used to analyze the data and followed by a posthoc test Tukey–Kramer pairwise comparison. *p* < 0.05 was indicated as statistically significant.

## 5. Conclusions

Enhanced LC3B-II and p62 show that impaired autophagy in CCI-induced neuropathic pain may result from blocking the late degradation phase. Increased LAMP2 further confirmed the late phase of lysosomal dysfunction and autophagosome accumulation. Loganin mitigated CCI-increased LC3B-II, p62, and LAMP2, suggesting that it can modulate the autophagic flux in CCI-induced pain and neuroinflammation. Loganin has also been shown to inhibit apoptosis and protect spinal neuronal cells (Figure 6). Thus, loganin may help provide neuroprotection and control of neuropathic pain.

## 6. Study Limitations

Autophagy is a highly dynamic event, and it is important to study its progress at different times after peripheral nerve injury. We decided to study the CCI rat model on day 7 since the most severe neuropathic pain occurred on this day. This study did not clarify whether loganin could stimulate autophagy through other potential molecular mechanisms in the CCI rat model. In-depth studies need to be conducted to investigate other potential signaling pathways of loganin in autophagic flux after CCI.

## Figures and Tables

**Figure 1 ijms-23-15873-f001:**
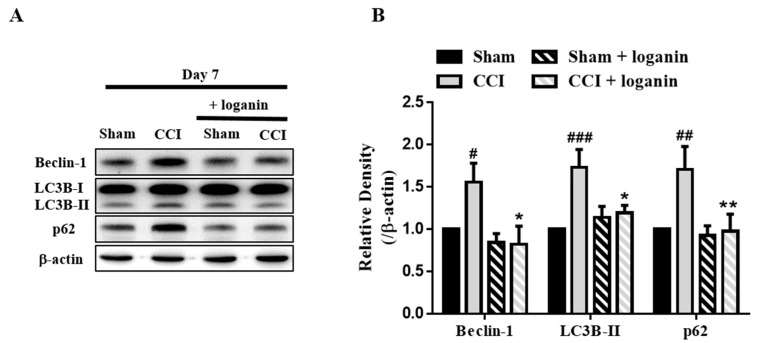
Effects of loganin (5 mg/kg/day, i.p.) on autophagy-related proteins, Beclin-1, LC3B-II, and p62 in the spinal cord on day 7 after chronic constriction injury (CCI). (**A**) Representative Western blot analysis of autophagy marker Beclin-1, LC3B-II, and p62 proteins. (**B**) Quantification of Western blot data in A. Sham-operated rats were the subjects of the same procedure without nerve manipulation. These data represent the mean ± SE for 6 rats per group. # *p* < 0.05, ## *p* < 0.01, ### *p* < 0.001 compared with the sham group at the corresponding time points; * *p* < 0.05, ** *p* < 0.01 compared with the CCI group.

**Figure 2 ijms-23-15873-f002:**
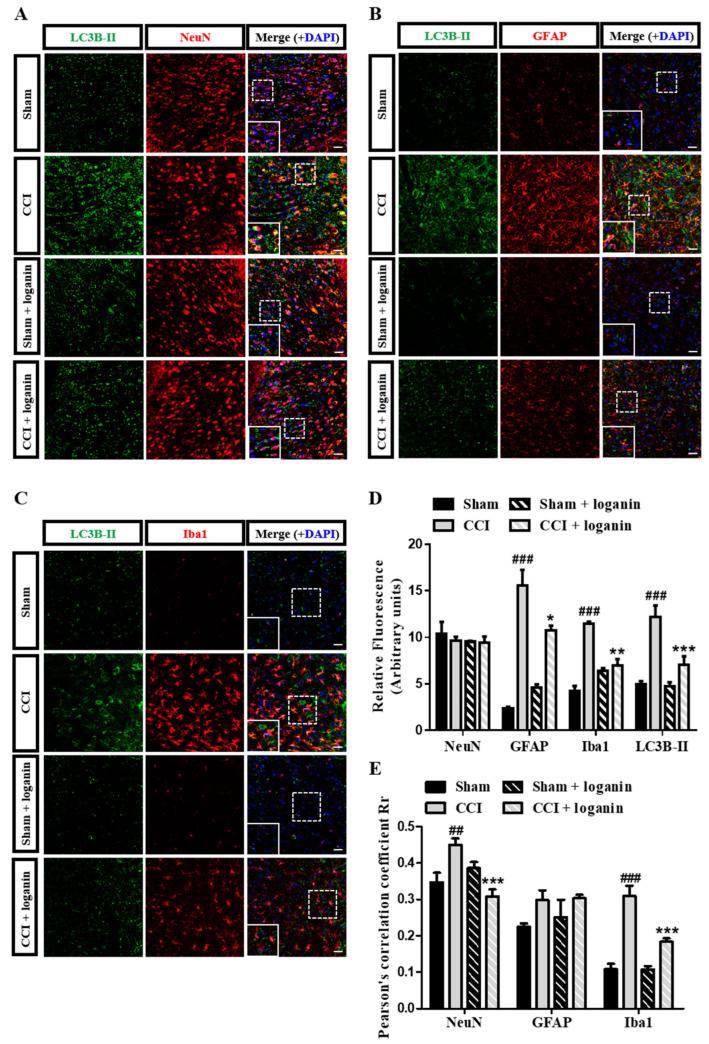
Double immunofluorescent staining for LC3B-II (autophagosome marker) and NeuN (neuronal marker), GFAP (astrocyte marker), and Iba1 (microglia marker) in the spinal cord on day 7 after chronic constriction injury (CCI). Representative images of IHC staining for LC3B-II (green) and (**A**) neuronal marker NeuN (red), (**B**) astrocyte marker GFAP (red), and (**C**) microglia marker Iba1 (red) in the spinal cord. The region of interest’s fluorescence (dash box) is amplified at the bottom box. The scale bar is 50 μm. (**D**) Quantification of NeuN, GFAP, Iba1, and LC3B-II signals in (**A**–**C**). (**E**) Pearson’s correlation coefficients quantified the degree of colocalization between LC3B-II and each neuron marker from (**A**–**C**). Sham-operated rats were the subjects of the same procedure without nerve manipulation. These data represent the mean ± SE for 6 rats per group. ## *p* < 0.01, ### *p* < 0.001 compared with sham group at the corresponding time points; * *p* < 0.05, ** *p* < 0.01, *** *p* < 0.001 compared with CCI group.

**Figure 3 ijms-23-15873-f003:**
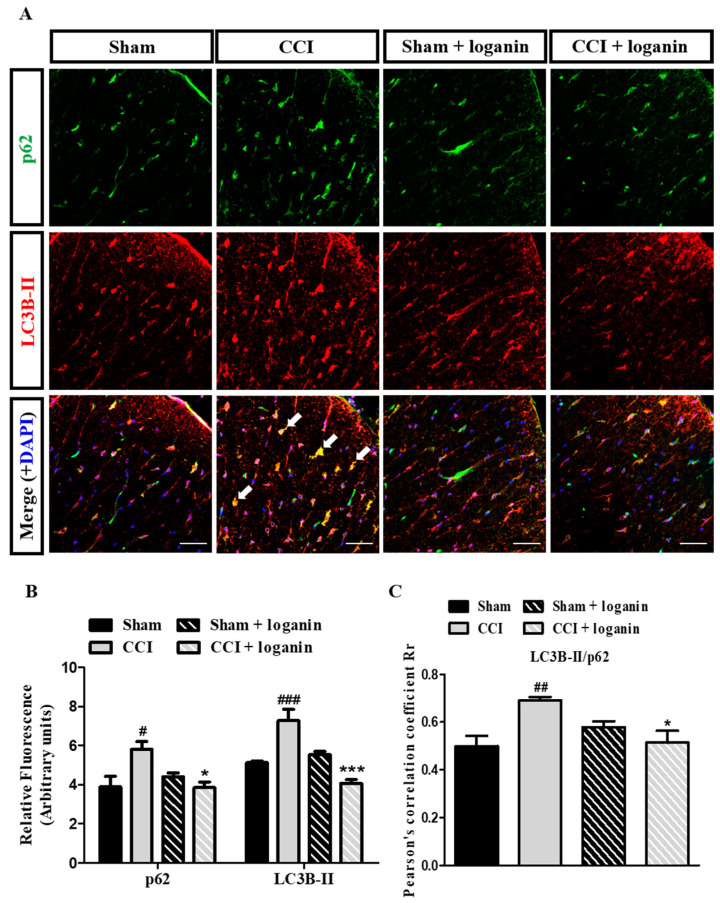
Double immunofluorescent staining for LC3B-II (autophagosome marker) and p62 (autophagic flux marker) in the spinal cord on day 7 after chronic constriction injury (CCI). (**A**) Representative images of IHC staining for p62 (green) and LC3B-II (red) in the spinal cord. Arrows point to representative LC3B-II and p62 colocalization. The scale bar is 50 μm. (**B**) Quantification of p62, LC3B-II signals in (**A**). (**C**) Pearson’s correlation coefficients quantified the degree of colocalization between p62 and LC3B-II. Sham-operated rats were the subjects of the same procedure without nerve manipulation. These data represent the mean ± SE for 6 rats per group. # *p* < 0.05, ## *p* < 0.01, ### *p* < 0.001 compared with sham group at the corresponding time points; * *p* < 0.05, *** *p* < 0.001 compared with CCI group.

**Figure 4 ijms-23-15873-f004:**
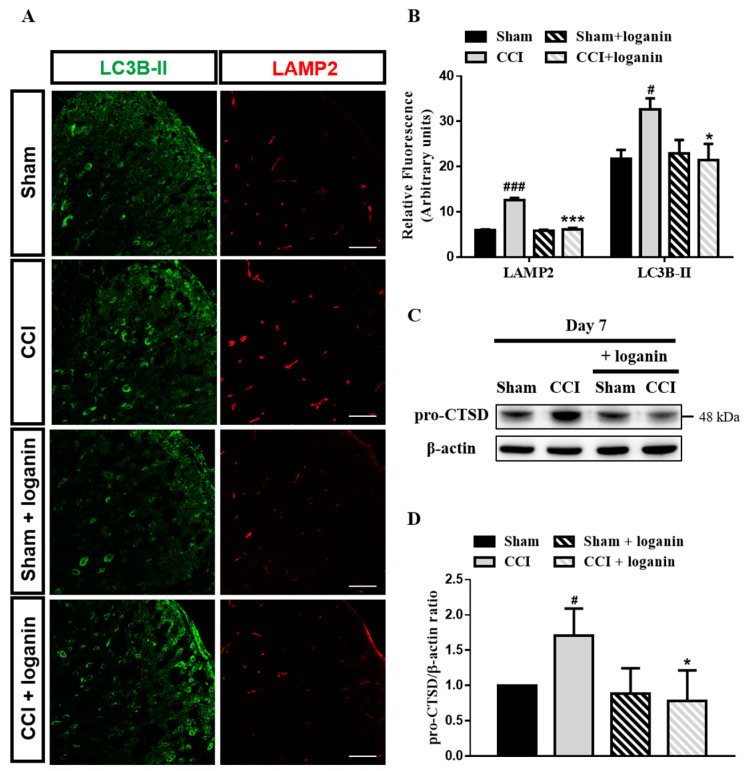
Effects of loganin (5 mg/kg/day, i.p.) on CCI-induced lysosomal dysfunction. (**A**) Representative images of IHC staining for autophagosome marker LC3B-II (green) and lysosome marker LAMP2 (red) in the spinal cord. The scale bar is 50 μm. (**B**) Quantification of LAMP2 and LC3B-II signals in (**A**). (**C**) Representative Western blot analysis of lysosomal protease Cathepsin D protein. (**D**) Quantification of Western blot data in (**C**). Sham-operated rats were the subjects of the same procedure without nerve manipulation. These data represent the mean ± SE for 6 rats per group. # *p* < 0.05, ### *p* < 0.001 compared with the sham group at the corresponding time points; * *p* < 0.05, *** *p* < 0.001 compared with the CCI group.

**Figure 5 ijms-23-15873-f005:**
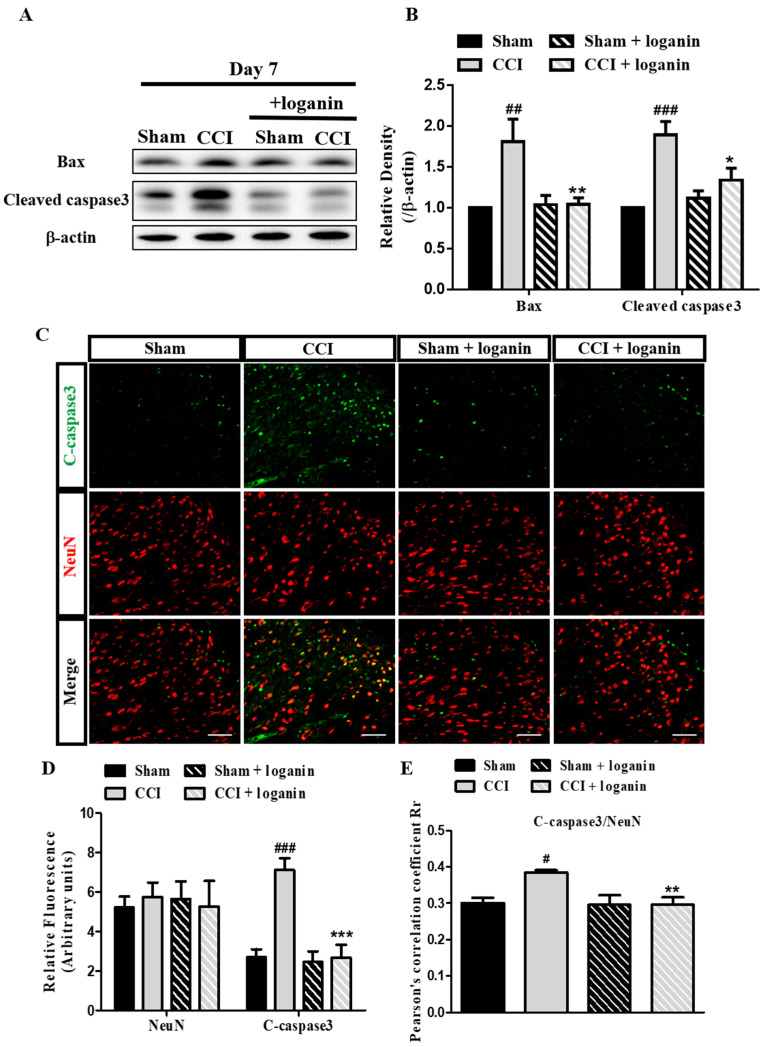
Effects of loganin (5 mg/kg/day, i.p.) on CCI impaired autophagic flux associated with neuronal cell death. (**A**) Representative Western blot analysis of Bax and cleaved caspase3 proteins. (**B**) Quantification of Western data in (**A**). (**C**) Representative images of IHC staining for C-caspase3 (green) and the neuronal marker NeuN (red) in the spinal cord. The scale bar is 50 μm. (**D**) Quantification of NeuN and C-caspase3 signals in (**C**). (**E**) Pearson’s correlation coefficients quantified the degree of colocalization between C-caspase3 and NeuN. Sham-operated rats were the subjects of the same procedure without nerve manipulation. These data represent the mean ± SE for 6 rats per group. # *p* < 0.05, ## *p* < 0.01, ### *p* < 0.001 compared with the sham group at the corresponding time points; * *p*< 0.05, ** *p* < 0.01, *** *p* < 0.001 compared with the CCI group.

**Figure 6 ijms-23-15873-f006:**
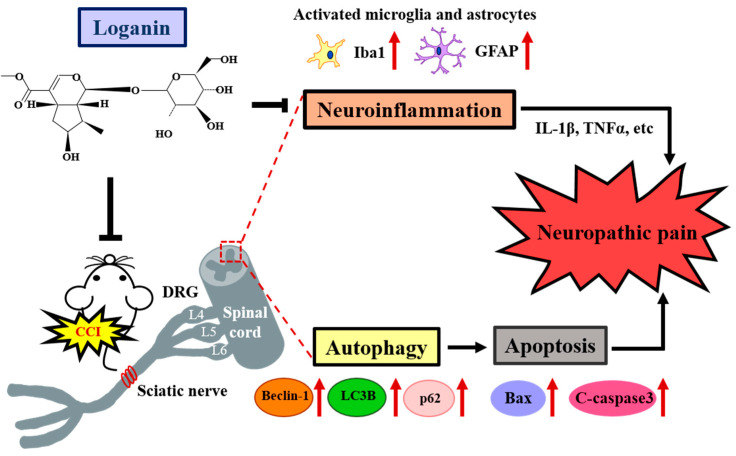
Proposed iridoid glycoside loganin mechanism in a CCI rat model. Loganin attenuates neuropathic pain by stimulating autophagy, inhibiting astrocyte/microglia excess activation, and reducing neuroinflammation and apoptosis.

## Data Availability

The corresponding author will provide the data supporting this study’s findings, which are available upon reasonable request.

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
