# Peer review of "The Iridoid Glycoside Loganin Modulates Autophagic Flux Following Chronic Constriction Injury-Induced Neuropathic Pain"

_ijms, 2022, doi:10.3390/ijms232415873_

Round 1

Reviewer 1 Report

Cheng et al reported the potential therapeutic benefit of loganin to the neuropathic pain by focusing the neuronal autophagy using a chronic constriction injury (CCI) rat model. Loganin’s analgesic and neuroprotective effects could be contributed by the regulation of autophagic flux in the CCI animal. This study was well-designed and carefully performed. The authors provided promising data to support the therapeutic effect of loganin. Nevertheless, following comments should be addressed before final acceptance.

 Major points:

1.     Does intraperitoneal loganin injection partially or completely reverse CCI-induced autophagy? Also, it is very critical to ensure if it reverses neuropathic pain behavior.

2.     In Fig. 1A, Beclin-1 level in CCI+loganin group is significant difference as compared to CCI group? Please check and clearly indicate in Fig.1B.

3.     The specific location(s) of the immune-positive cells in IHC image needs to be clarify. In addition, please improve the resolution of Fig.2!

4.     The legend of Fig. 3 describes “Arrows point to representative LC3B-II and p62 colocalization” ? Yet, the arrows are missing!

5.     In Fig. 5A, why the cleaved caspase 3 is abundant in sham+loganin group? Please check the data!

Author Response

Response to Reviewer #1

The authors appreciate your time and effort in reviewing our paper and giving us some helpful suggestions. As follows, we respond to your concerns point-to-point.

Major points:

  1. As you are concerned, we have removed the word partially.

Lines: 44-46 (in Abstract), Loganin prevents CCI-induced neuropathic pain, which could be attributed to the regulation of neuroinflammation, neuronal autophagy, and associated cell death. These data suggest autophagy could be a potential target for preventing neuropathic pain.

Lines: 259-261 (in Discussion), Our findings extend the past literature by demonstrating that loganin alleviated neuropathic pain after CCI, restoring autophagy flux.

  1. In Fig. 1A, we have re-analyzed the data; thanks for your comments. The Beclin-1 level in CCI+loganin group did significantly different compared to the CCI group. We have fixed this problem. Please see page 3 for details.
  1. Thank you for your suggestions. The region of interest fluorescences (immune-positive cells) is amplified at the bottom of figure 2. And the resolution is also improved.
  1. We have added the arrows in figure 3. Please see page 5 for details.
  1. In Fig. 5A, We have re-analyzed the data of cleaved caspase 3 in the sham+loganin group. The matter we have fixed. Please see page 7 for details.

Reviewer 2 Report

This submitted manuscript explored whether loganin could increase autophagy flux which mitigates CCI-induced neuronal death and neuropathic pain.  This is a continued work following their previous publication providing a more in-depth mechanism which underly loganin’s beneficial effects.  In general, this project has solid research background and the manuscript has been written in good structure, easy to follow and understand.  The results were clear and solid, and the conclusion was reasonable.  Also, the authors pointed out the limitation of this manuscript not overstating its significance.  No major concerns have been raised for this manuscript. 

Minor concerns:

1) The authors claimed that CCI could disrupt autophagy flux by decreasing lysosome function and loganin could restore the dysfunction.  The expression of LAMP2 is an index for lysosomal numbers and indirectly represents lysosomal function.  The authors should investigate the expression levels of cathepsins such as cathepsins B and D as wells as lysosomal damage marker galectin 3.  With these data, the conclusion should be more convincing.   

2) In a couple of places, the authors wrote LC3 instead of LC3B (Line 144, 147).  These should be corrected. 

Author Response

Response to Reviewer #2

The authors want to thank the reviewer's positive support for our work. Our point-to-point responses are listed below.

Minor concerns:

  1. Thank you for your insightful suggestions. We have added the pro-cathepsin D (pro-CTSD) data in Figures 4C & 4D. The interpretation of such data is also added on page 6, lines 170-174. Please see page 6 for details. Thanks again.
  2. We have fixed and highlighted the word LC3B (Now, lines 145 &158); thanks.  

Round 2

Reviewer 1 Report

All of my comments have been well addressed, No more comment!

I am pleased to accept this paper, Congrats!